

# Impact of green technology innovation based on IoT and industrial supply chain on the promotion of enterprise digital economy

Ruilin Song[1,2] and Hui Hu[1,2]

[1] Economics and Management Division, Wuhan City College, Wuhan, Hube, China
[2] Hubei Science and Technology Innovation High Quality Development Research Center, Wuhan, Hubei, China

## ABSTRACT

With the gradual deterioration of the natural environment, a green economy has become a competing goal for all countries. As a trend of green innovation development, the digital economy has become a research hotspot for scientists. In this article, we study the supply chain management of enterprises in green innovation and digital economy development and complete the identification and demand prediction of warehouse goods through the Internet of Things (IoT) and artificial intelligence (AI). As the stuff meets the goods detection and storage, we employ an intelligent method to detect and classify the goods. The demand prediction analysis is carried out based on historical data on goods demand in the enterprise. The absolute error between the prediction result and the actual demand within 1 week is less than 30 goods by the particle swarm optimization-support vector machine (PSO-SVM) method used in this article. First, the goods identification task is completed based on video surveillance data using YOLOv4, and the recognition rate is as high as 98.3%. This article realises enterprises' intelligent supply chain management through the intelligent identification of goods and the demand forecasting analysis of goods in the warehouse, which provides new ideas for green innovation and digital economy development.

## INTRODUCTION

Today's society is developing rapidly based on the rapid consumption of resources and environmental pollution and destruction, and the contradiction between man and nature is deteriorating (*Oduro, Maccario & De Nisco, 2021*). As a result of this enormous challenge, people have gradually begun to understand the significance of green environmental protection, strengthened energy conservation and emission reduction, promoted green environmental protection, resource recycling has gained widespread acceptance, and some manufacturing enterprises have started devoting themselves to the development and production of green products with environmental performance to increase marketing share and product competitiveness. In this context, the green supply chain has emerged, providing a new operating mode for enterprises. The adoption of green

Corresponding author
Ruilin Song,
song_1301419@163.com

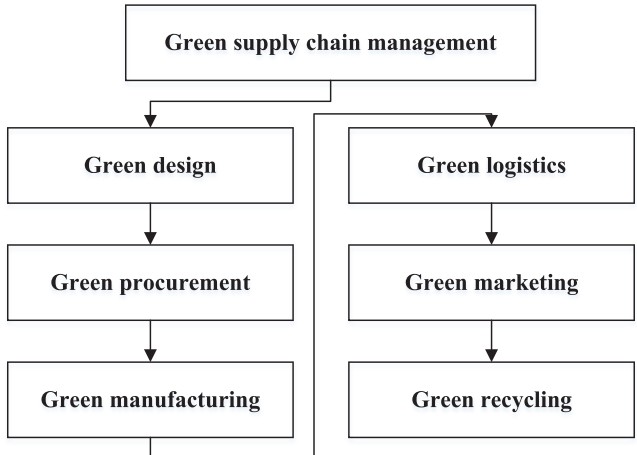

**Figure 1** The management for the green supply chain management.

supply chain management contributes significantly to the long-term viability of businesses and serves as an efficient operational plan for the growth of contemporary businesses. Enhancing the efficiency of green innovation that considers "green" and "innovation" becomes the key to realising the development of green transformation in the region. The management of a green supply chain can be summarised into six main links, as shown in Fig. 1 (*Tseng et al., 2019*).

The information shown in Fig. 1 shows that the overall development and management of the green supply chain allows for a high level of science and technology to achieve. With the promotion of economic globalisation and the rapid growth of IoT, the demand range of enterprises is widely expanded through traditional and online sales channels. The involvement of e-commerce has heated the competition among companies, which must constantly adopt strategies that focus on adapting product features to meet customer needs and preferences. Many companies are committed to improving logistics services such as warehousing and distribution to attract consumers. The high level of logistics services has increased inventory costs on business days. IoT technology has had a significant impact on supply chain development. Following are some of the ways it has transformed the supply chain: First, real-time tracking: IoT devices can track goods and materials in real-time, providing businesses greater visibility and control over their supply chain. Secondly, predictive maintenance: IoT sensors can monitor the condition of equipment and machinery, enabling enterprises to perform predictive maintenance and avoid downtime. Thirdly, inventory management: IoT devices can monitor inventory levels and automatically reorder supplies when stock levels fall below a certain threshold. At last, increased transparency: IoT technology can provide greater transparency and traceability throughout the supply chain, enabling businesses to identify and address issues quickly. Overall, IoT technology has the potential to revolutionise supply chain management by improving efficiency, reducing costs, and increasing transparency and visibility.

In addition, the application of advanced data collection systems allows enterprises to collect and transmit data from almost all processes. Enterprises face exponential data

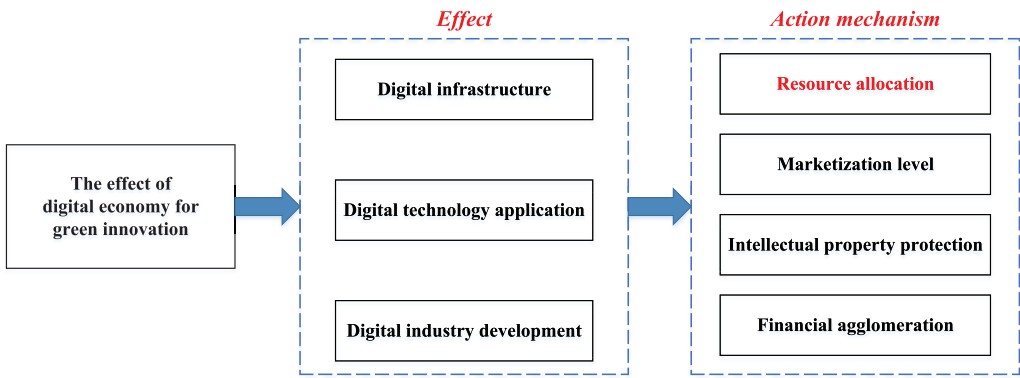

**Figure 2 The relationship between digital economy and green innovation.**

growth, and demand increasingly shows greater volatility and randomness. Businesses are dealing with an exponential increase in data, and demand is becoming more volatile and erratic. Enterprises cannot effectively exploit the vast amounts of supply chain data available, making it difficult for them to adapt rapidly to changes in demand. As a result, they keep inventory levels high to prevent stockouts, which also raises the cost of inventory (*Haiyun et al., 2021*). Promoting computer technologies, such as big data and machine learning, has brought new opportunities for companies. Machine learning can derive information from data, make decisions based on changes in environmental factors, and continuously improve performance as data is input. Therefore, using machine learning algorithms to solve enterprise inventory management problems has become a viable option for companies (*Benzidia, Makaoui & Bentahar, 2021*).

With the widespread application of IoT and AI, the digital economy has also been developing rapidly. Numerous new technologies and business models have emerged due to the close relationship between the real world and the digital economy. These innovations have altered the process of creating value and given rise to fresh approaches to resolving the tension between traditional economic growth and environmental protection and fulfilling the value demand for the synergistic growth of economic and environmental benefits (*Pan et al., 2022*). The effect of the digital economy on innovation is shown in Fig. 2.

In Fig. 2, the digital economy contributes to green innovation through its impact in three areas: digital infrastructure, technology application and industry development. Optimizing resource allocation is the most important mechanism in which the digital economy influences green innovation. It is easy to see that green innovation and the digital economy complement each other, and AI and IoT technologies are essential mediums to connect the two. Therefore, combining artificial intelligence and IoT technology is vital for the digital economy and enterprises' strategic development by improving the supply chain's resource allocation and green scheduling (*Litvinenko, 2020*).

Green innovation is pursued by various enterprises as the focus of current economic development, and IoT technology is an essential means to achieve green innovation. Therefore, this article studies the supply chain inventory problem in digital economy

development and completes the supply chain management through machine learning technology to improve the efficiency of enterprise operations and realize green innovation. The specific contributions are as follows.

(1) This article builds an inventory goods recognition model based on the YOLOv4 framework and completes the automatic identification of goods in the warehouse.
(2) Forecasting inventory demand in an enterprise's supply chain based on the PSO-SVM method, achieving the goal of planning enterprise strategies.
(3) After completing the model construction, this article performs goods identification and demand forecasting for the enterprise's near-term inventory based on the proposed framework. The results are consistent with the strategic sector budget.

The rest of the article is organised as follows: "Related works" introduces the related works for digital economy development and supply chain optimization; "Machine learning-based enterprise demand and inventory cost analysis model construction" presents the methods employed. "Experiment result and analysis" describes the experiment and results from the analysis of the proposed framework, and the practical test for the framework is also given. "Discussion" discusses the result and the notice that should be paid in the development of digital economy. The conclusion is drawn in the "Conclusion".

# RELATED WORKS

## Corporate digital innovation research

Most scholars do not have a unified understanding of digital innovation in manufacturing. Most give different explanations based on different perspectives, mainly based on the three levels of the innovation process, the outcome and the integrated study of both. At the level of digital innovation results, *Sahut, Dana & Laroche (2020)* consider digital innovation as the recombination of digital and physical components to produce new products, which implies value creation and performance improvement through digital technology and business model innovation. At the level of digital innovation process, *Lyytinen, Yoo & Boland (2016)* on the other hand, consider digital technologies and tools as operational resources and embed them in the innovation process through digital connectivity and digital convergence. *Nambisan et al. (2017)* argue that digital technologies and digital tools give rise to digital innovation, that the creation of new products and innovative business processes through digital technology is the essence of digital innovation, and that the results of digital innovation can be achieved without digitization, as long as the innovation process directly or indirectly utilises digital technology. *Leydesdorff & Ivanova (2016)* argue that the evolution of the open innovation model in the digital era is no longer exactly like traditional innovation in general, and *Oborn et al. (2019)* argue that innovation in the digital era is more likely to occur through the spatial and temporal distribution of emerging heterogeneous networks, forming a dynamic process of feedback loops. In emerging heterogeneous networks, developing a platform innovation model and innovation will become more uncertain and complex due to the heterogeneity and uncertainty of

innovation agents in the network platform (*Oborn et al., 2019*). Digital platforms created by digital technologies blur the boundaries of innovation processes and outcomes and encourage the growth of distributed and combinatorial innovation, according to *Yoo, Henfridsson & Lyytinen (2010)*, who argue that digital technologies will change the nature of products and services and that digital innovation is essentially a reorganization of digital technology at the integrated consideration of digital innovation outcomes and processes.

The current digital economy is mainly a forward-looking study of technology's development and application scope based on the above innovation process and results. With the continuous application of IoT, the above ideas are realised through artificial intelligence and other methods. Applying cutting-edge technology to business development can encourage and support the digital economy.

## Enterprise supply chain optimization research

Inventory optimization is using inventory control to reduce a company's inventory costs, which is crucial to reducing the capital tied up in the company by lowering inventory costs. Inventory control involves the issues of when to restock raw materials and goods, determining the restocking amount, and safety stock. Gunasekaran et al. used three different MCDM methods for ABC analysis to determine the appropriate product for each inventory category. They applied it to inventory management at a large automotive company (*Kartal et al., 2016*). *Sitompul et al. (2008)* studied the relationship between demand variability, capacity constraints, and safety stock and proposed an alternative model for setting up safety stock in a supply chain with a sufficient capacity. *Galli et al. (2021)* proposed an inventory optimization model for the medical inventory domain to minimise inventory levels and the need for emergency replenishment. *Duarte, Gonçalves & Santos (2021)* developed a mixed integer linear programming model and solved it with a solver based on a branch-and-cut algorithm. Finding the optimal production and inventory strategy in a multi-product baking unit was solved. System dynamics models have also been used to solve inventory problems due to the complex dynamics of supply chain multilevel inventories, such as nonlinearity, dynamics, and delays. *Poles (2013)* used a system dynamics simulation modelling approach to model the production and inventory system for remanufacturing. They explored the dynamic remanufacturing process to give inventory and production improvement strategies for the simulated system. Making decisions for uncertainty is an essential issue in supply chains, where the processes in the flow of supply chain goods and services contain many complex decision processes and information barriers. To improve supply chain visibility, many leading organizations share information from both ends of their supply chains with their suppliers, and supply chain management is becoming increasingly data intensive (*Ali et al., 2017*). Realising the growing importance of data in supply chains, supply chain practitioners have tried every possible way to improve them. Machine learning is considered one of the methods. Machine learning algorithms have become a new hot spot for supply chain optimization solutions research due to their nonlinear solid data processing capabilities and predictive capabilities in unsupervised domains. *Cavalcante et al. (2019)* studied different machine learning algorithms in supplier selection, combining machine learning and simulation

techniques to improve supplier delivery reliability. *Priore et al. (2019)* used a random forest algorithm to determine the optimal replenishment strategy for different commodities in a three-level supply chain to reduce the supply chain bullwhip effect. Wang et al. considered the effects of power optimization and cost factors. They designed and developed a statistical cost-based approach based on multiple costing frameworks using machine learning models (SCM-MLM) (*Wang & Zhang, 2020*). *Martínez et al. (2020)* developed a machine learning framework for predicting whether a company's customers will buy again in the future. *Huber et al. (2019)* proposed a data-driven machine learning and quantile regression-based prediction method and applied it to point-of-sale data from a large German bakery chain.

It is clear from the study of supply chain optimization and the growth of the corporate digital economy that various industries are investigating using artificial intelligence (AI) technology throughout the whole cycle of their enterprise operation to increase efficiency. The application of a large amount of data also brings the data economy come into being. Scheduling for the corporate supply chain provides inventory optimization, identifies commodity products while optimizing inventory, and allows the supply chain to play a part in the enterprise's digital economy. Only accurate scheduling can achieve green development.

## MACHINE LEARNING-BASED ENTERPRISE DEMAND AND INVENTORY COST ANALYSIS MODEL CONSTRUCTION

### Framework establishment of Yolov4-based warehouse goods identification analysis

Convolutional neural network (CNN) is the primary tool for processing image information. The training process of CNN models is learning the relationships between image pixels, which are reflected in the model parameters through convolutional operations. The convolution process of 3D convolutional neural networks is shown in Eq. (1).

$$f_{out}(v_{ti}) = \sum_{v_{ti} \in B(v_{ti})} \frac{1}{Z} fea_{in}(v_{ti}) \cdot \omega(l_{ti}(v_{ti})) \tag{1}$$

The 3D convolutional neural network's unique kernel structure can learn temporal and spatial dimensions data. 3D convolutional neural networks can be trained and used immediately using raw video data as the network's input, removing the requirement to first remove optical flow and video frame data. Inventory management, sensitive to time information, is one application where 3D neural networks are highly successful at detecting content.

The YOLO target detection algorithm is a family of efficient single-step target detection algorithms proposed by *Redmon & Farhadi (2018)*. The core idea is to perform the bounding box regression and identify the category to which the input image belongs directly in the output layer. With the continuous iteration of the algorithm, the current most popular one is the YOLOv4 method. YOLOv4 has various applications in various

industries, such as retail, transportation, and security. YOLOv4 can be used in retail for product recognition, stock management, and customer analysis. For example, it can be used to track the movement of products in a store, analyse customer behaviour, and detect product placement errors. YOLOv4 can be used in transportation for autonomous driving, traffic management, and vehicle tracking. It can detect objects such as pedestrians, vehicles, and traffic signs and provide real-time information to autonomous vehicles. It can also be used to monitor traffic flow and notice accidents. In security, YOLOv4 can be used for surveillance, crowd monitoring, and facial recognition. It can detect suspicious activity, monitor crowds, and identify individuals. It can also be used to track the movement of people and vehicles in restricted areas. Overall, YOLOv4 is a powerful and versatile object detection algorithm that can be used in various applications. It's high accuracy and speed make it an ideal choice for real-time applications that require fast and reliable object detection.

This version firstly incorporates the cross stage partial network (CSPNet) based on the YOLOv3 backbone network Darknet, which reduces memory consumption. Secondly, YOLOv4 uses the Mish activation function to reduce the computational cost. The expression of the Mishap function is shown in Eq. (2).

$$Mish = x\tanh(\ln(1 + e^x))  \qquad (2)$$

It can be seen that the Mish activation function does not take an utterly truncated value at negative values, it allows the inflow of negative values with relatively small gradients, and the Mish function is unbounded by this feature, avoiding the problem of gradient saturation (at the limit position, the gradient tends to 1). At the same time, the curve of this function is smooth and continuous compared with the traditional Relu function, and the derivative is better, which helps to improve the localization effect on the target. Besides, adding the Squeeze-and-Excitation Net (SENet) (*Hu, Shen & Sun, 2018*) to the deep network adds an attention module to the channel feature map to enhance the target's weight. Finally, the CSPDarknet53 is used as the backbone network. YOLOv4 neck uses the Path Aggregation Network (PANet) with the Spatial Pyramid Pooling (SPP) layer instead of the original Feature Pyramid Network (FPN). This module alleviates the problem of losing the location and contour information of the target to a certain extent. Based on the above advantages, this method improves the target detection accuracy based on YOLOv3.

## PSO-SVM-based enterprise demand forecasting

For enterprise demand, its inventory changes according to specific patterns, and these patterns are related to various factors such as season, economic level, *etc.* The optimal way to solve such problems is to complete demand forecasting based on historical data to explore the data patterns (*Huang et al., 2021*). This article uses the more classical SVM method in machine learning to complete enterprise demand forecasting. Support vector machines (SVM) is a supervised learning method, and this model can show strong processing ability and high performance in dealing with some data with high dimensionality. For the SVM model, the core of its algorithm is to find the optimal

hyperplane, which is to find the parameter that satisfies the distance from the interval sample points to the hyperplane and the maximum $\omega$ and $b$. The classifier corresponding to the decision boundary that maximizes the interval is a hard interval support vector machine. Maximising the interval becomes a problem of minimizing $\omega$ The minimization problem, according to its principle, *i.e.*, to select the optimal $\omega$ and, we can obtain the loss function as shown in Eqs. (3) and (4).

$$\min_{\omega,b} \frac{1}{2}\omega^2 \tag{3}$$

$$\text{s.t. } 1 - y_i(\omega^T x_i + b) \leq 0, \ i = 1, 2, \dots, n \tag{4}$$

The optimization problem represented by the above two functions is a convex quadratic programming problem. By adding Lagrangian multipliers to each constraint $\alpha = (\alpha_1, \alpha_2, \dots, \alpha_m)^T, \alpha_i \geq 0$, the objective function is transformed to solve $L(\omega, b, \alpha)$ the extreme value problem, concerning $\omega$, b solving for the minimum values and the $\alpha$ solving for extreme values. In practical applications, not all issues are linearly differentiable, so a soft interval is needed for error correction, *i.e.*, the introduction of $\xi_i$ Relaxation variables enhances the model capability, and the original problem after the introduction of relaxation variables can be expressed by Eqs. (5) and (6).

$$\min_{\omega,b,i} \frac{1}{2}\omega^2 + c\sum_{i=1}^{n} \xi_i \tag{5}$$

$$\text{s.t.} \begin{cases} y_i(\omega^T x_i + b) \geq 1 - \xi_i, i = 1, 2, \dots, n \\ \xi_i \geq 0, i = 1, 2, \dots, n \end{cases} \tag{6}$$

When using the SVM method to complete enterprise demand forecasting based on historical data, we discovered that the SVM method is easily prone to fall into the local optimum, making it challenging to achieve model optimization. For this reason, this article chose the particle swarm method to optimize the model. PSO is a search algorithm with few parameters and is easy to implement, which searches for the optimal solution by iteration. The main steps of its algorithm are as follows (*Zhao & Zhao, 2021*):

To represent the process of PSO optimization SVM more visually, the algorithm process update is shown in Fig. 3.

By optimizing the parameters of SVM, we achieve efficient fitting based on historical inventory data from warehouses in the supply chain to complete inventory forecasting.

# EXPERIMENT RESULT AND ANALYSIS

## The recognition of the goods in the warehouse

The current application of IoT in warehouse management of the supply chain is mainly video monitoring technology, so this article selects the highest-selling product in this warehouse in the last month for identification. Because of the enormous sales volume of this product, its mobility is fast, which increases the difficulty of model training. In the model construction, the period of the item appearing under the fixed camera during the

| PSO Algorithm |
| --- |

*Step1:* Parameters initialization including the particle populations, velocity and position of each particle.

*Step2:* Fitness calculation using the fitness function

*Step3:* Compare the Pbest with the fitness; if Fitness>Pbest, then Pbest=Fitness

*Step4:* CompareGbest with fitness; if Fitness>Gbest, then Gbest=Fitness.

*Step5:* Update the speed and position according to Eqs. (7) and (8):

$$v_i(t+1) = \omega v_i(t) + r_1 c_1 \left( \text{Pbest}_i - \text{pop}_i(t) \right) + r_2 c_2 \left( \text{Pgest}_i - \text{pop}_i(t) \right) \tag{7}$$

$$\text{pop}_{i(t+1)} = \text{pop}_i(t) + v_i(t+1) \tag{8}$$

where $\omega$ is the inertia factor, $r_1, r_2$ are two random numbers ranging $[0,1]$ between, $c_1, c_2$ is the learning factor, which denotes the ability of the particle to accelerate to its own historical optimal solution and the ability of the particle to accelerate to the current global optimal solution, respectively.

*Step6*: Termination judgement: max iteration and best fitness.

working period of a week is selected for data annotation to complete the model training. Meanwhile, to accurately illustrate the model's performance, three indicators, Precision, Recall and F1-score are chosen to evaluate the model.

This study uses 30% of the data as the test, and the standard data division approach is chosen for model validation. The experimental results obtained are shown in Fig. 4.

YOLO, the most widely used method in target detection research, has a significantly higher recognition rate than its predecessor methods. As can be seen in Fig. 4, the precision is as high as 98.3% when YOLOv4 performs target recognition. It is significantly higher than conventional machine learning methods and CNN methods.

## The prediction for the demand the goods

To complete the analysis of the forecasted commodity demand for this year, this article combines historical data from the company's warehouse management process over the previous 5 years. To achieve intelligent warehouse management, the outcomes of commodity demand forecasting under various methodologies are shown in Fig. 5.

According to the results shown in Fig. 5, it can be found that due to the sufficient amount of data and the considerable random seasonal variation of commodities, all types of methods can forecast the commodity demand trend with a slight difference in the main trends. However, a more detailed comparison shows that the PSO-SVM method has better forecasting accuracy. The absolute forecast errors of different techniques in different months are shown in Fig. 6, while the sum of forecast errors for each month is shown in Fig. 7.

According to Figs. 6 and 7, we can see that the PSO-SVM method used in this article performs well in the prediction results. The SVM method in this article will have certain singularities in the prediction process, such as the April and June data in Fig. 6, both of

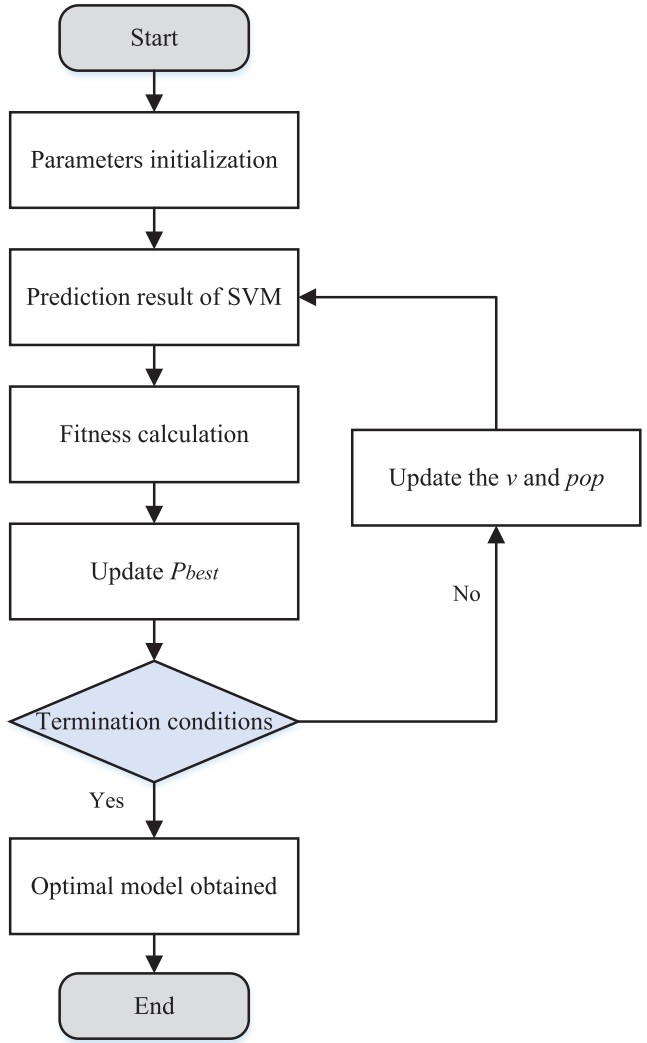

**Figure 3** The flow chart of PSO optimization SVM.

which have large deviations, and after optimization by the PSO method, the performance of the model is greatly improved, and better results are achieved.

## The framework application in practice

To test effectiveness of the method proposed in the practical application, practical test experiments were conducted in this article. The test framework is shown in Fig. 8.

In the actual testing process, we tested the model recognition rate of the used YOLOv4. We selected a warehouse working peak day within the video surveillance data for testing, and the comparison process is shown on the left side of Fig. 8. The statistical data of the duty officer was compared with the model recognition data based on the actual records of the warehouse personnel to form the true value. The results of the test are shown in Table 1.

It can be seen that the proposed method recognises higher data than manual counting in this camera view, which is because mental inattention often occurs in the manual counting

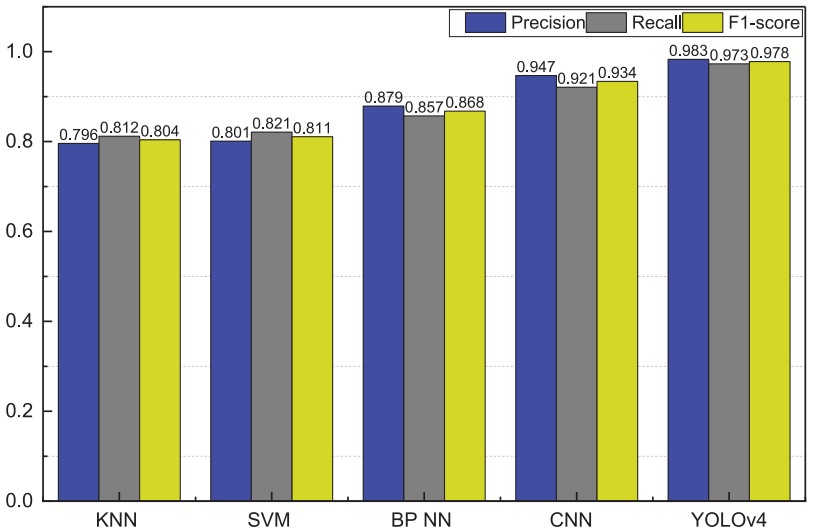

**Figure 4** The result of the goods recognition using the video surveillance data.

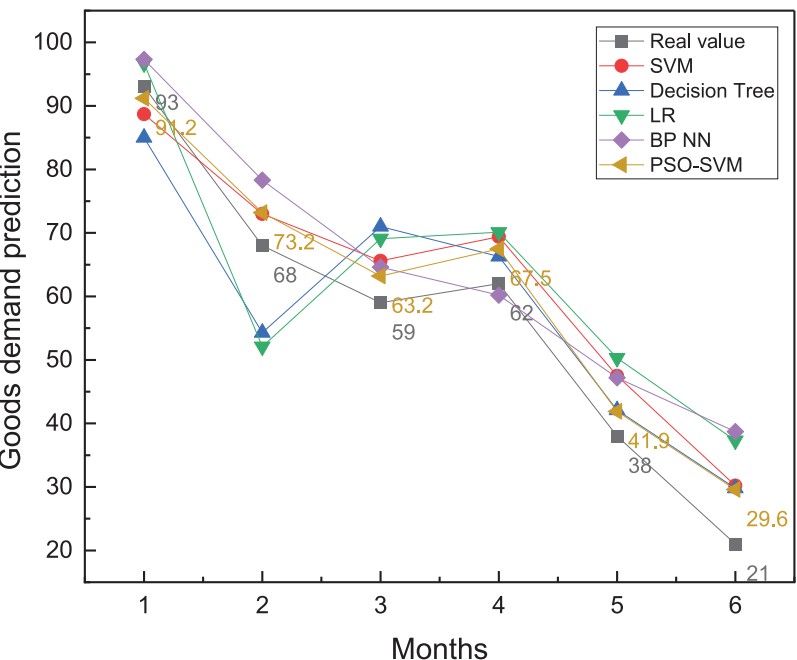

**Figure 5** The prediction result among different months.

process, leading to errors in data statistics. Therefore, the YOLO method used in this article is an excellent way to avoid such problems and improve the accuracy of goods recognition.

At the same time, this article forecasted the demand for goods in the latest month and compared it with the value set by the company's strategy department. In the latest month, the actual need for goods was 47 pieces, while the company's strategy department was forecast. The forecast of the proposed method was 41.2, rounding the data to 41 pieces, and

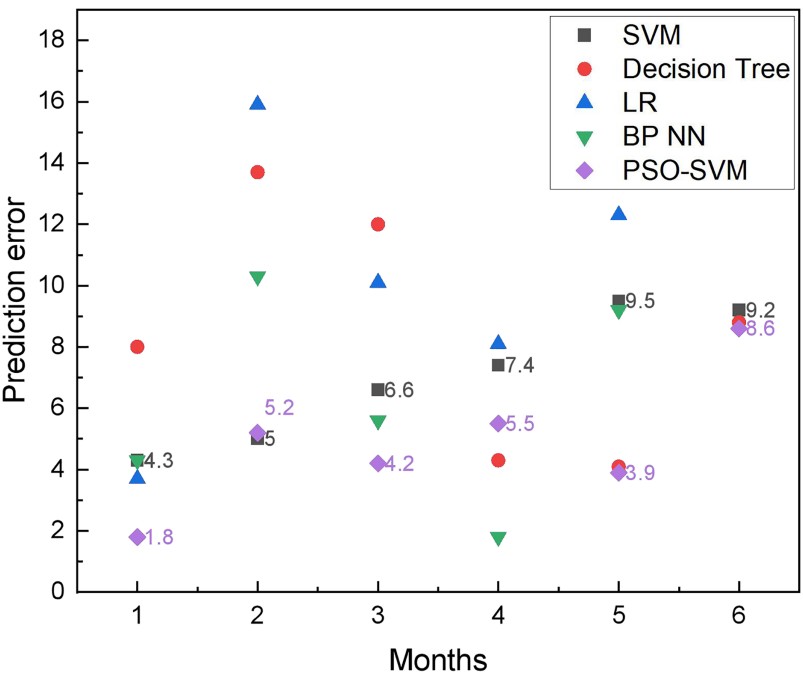

**Figure 6 The prediction error among different months.**

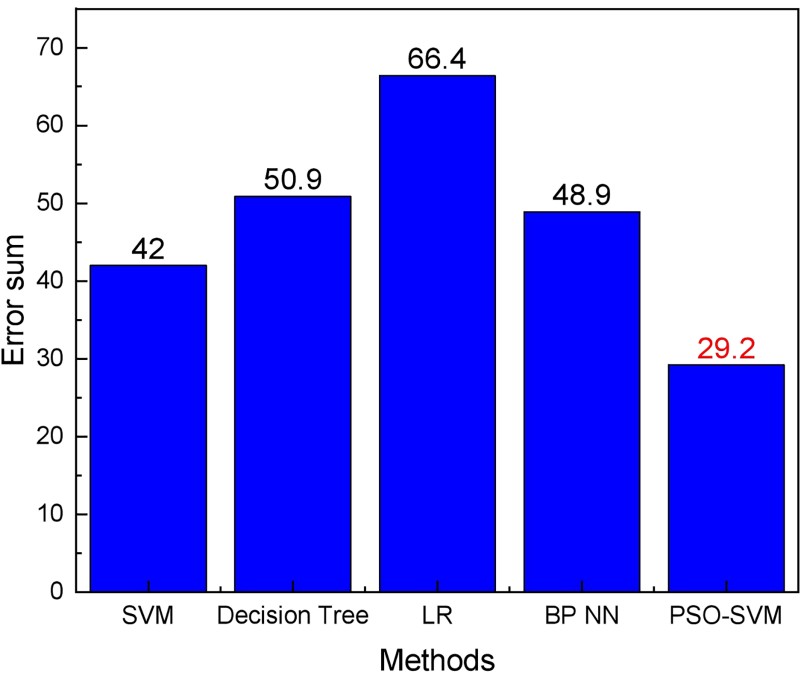

**Figure 7 The error sum of different methods.**

the difference between the two performances was insignificant. However, during the forecasting process, the strategy department researches various information, such as relevant policies and consumption habits, in detail. This process requires a large amount of human and material resources. The proposed PSO-SVM method completes the demand

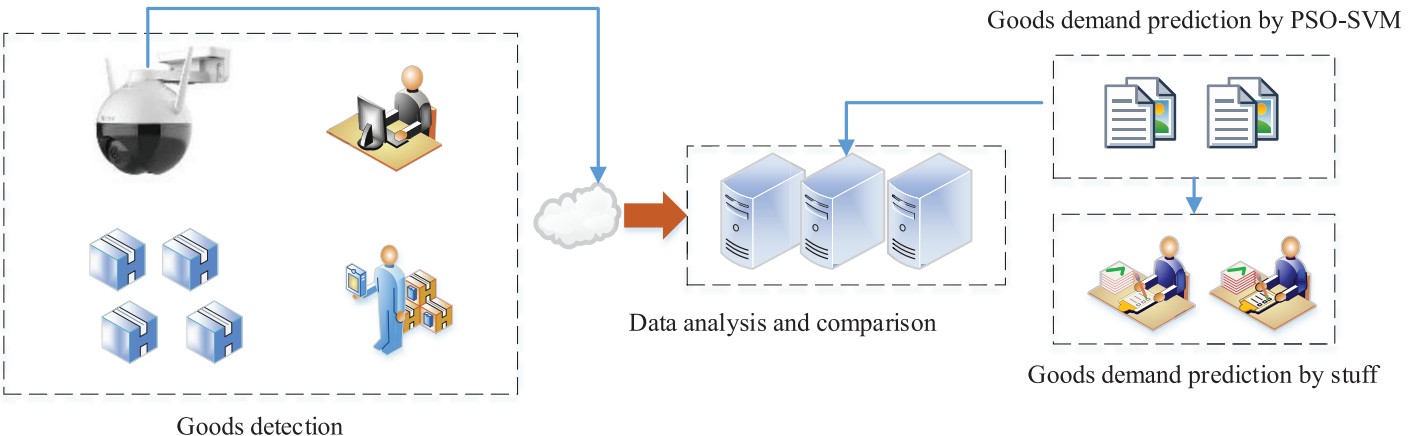

**Goods demand prediction by PSO-SVM**

Data analysis and comparison

**Goods demand prediction by stuff**

Goods detection

**Figure 8** **The framework for the model test.**

**Table 1** **The result for the goods detection.**

| Methods | Number |
| --- | --- |
| Warehouse statistics | 21 |
| YOLO detection | 20 |
| Artificial counts | 17 |

forecast based on historical data, an essential reference for future policy formulation and strategy adjustment in the strategy department.

## DISCUSSION

In today's high-speed development of IoT, the essence of saving labour costs through high-tech means is a green innovation. In this study, AI is utilised to finish the intelligent transformation of inventory management in the supply chain of firms, following the needs of businesses and the law of economic development. YOLO, as a classical target detection algorithm framework, combines the anchor box of Faster R-CNN with the groundbreaking multi-resolution prediction to achieve a swift and accurate detection effect (*Diwan, Anirudh & Tembhurne, 2022*). YOLOv4 further improves the performance of the model by enhancing its activation function. In demand prediction of goods, this article uses machine learning methods to fully exploit the information on goods in the past years and build a prediction model. In the model prediction, the deep learning method is not chosen because, for such data with a small amount and simple structure, the deep learning method will consume a lot of computational resources and cannot meet the essence of green innovation. Therefore, in this article, the classic SVM method is chosen to build the model, and the PSO method with lower computational complexity is selected to complete the parameter optimization in the process of model optimization. In the comparison of the results, it is found that for the predicted 6-month data, the absolute error of the proposed method is 29.2, which outperforms other approaches.

Green innovation drives the development of the digital economy, and technical research around how to use and maintain data will become a hot research topic in the coming period. Overall, the digital economy can significantly improve the efficiency of regional green innovation, which is conducive to practising the new development concept, driving the transformation of the economy to green development, and thus achieving high-quality economic development. Therefore, as the global technological revolution progresses, government departments in various regions must enhance the infrastructure for supporting digital technologies by speeding up the development of 5G and other IoT technologies' digital infrastructure, reducing the gap in digital construction, expanding the reach of digital technologies across regional boundaries, and laying a solid material foundation for the distribution of digital dividends (*Tian et al., 2021*). Second, investment in research and development of digital application scenarios in the industry should be increased to broaden the scope of digital technology applications while building a platform for integrating digital technology and industry so that digital technology can better empower technological innovation in other sectors. Again, the government leads social capital to participate in the transformation of local economic structure, increases investment in the field of digital technology basic research, drives traditional industries to gradually transform to digital industrialization and industrial digitalization, promotes the development of core industries of the digital economy, and releases the potential of data elements (*Ma & Zhu, 2022*). Green innovation and the data economy complement each other. At the same time, technologies such as IoT and AI, as intermediate media, need to be used in more application scenarios to release the potential of such technologies. In this article, we provide a management framework that aligns with both the concept of green innovation and the development of the digital economy for the inventory problem of corporate supply chains, intending to develop new ideas for future business development.

## CONCLUSION

This article researches the problem of inventory wisdom management in the enterprise supply chain in the context of green innovation and the digital economy. First, it proposes a warehouse goods identification method based on IoT monitoring data which uses YOLOv4 as a framework and achieves a 98.7% identification rate for items under a fixed monitoring perspective. Second, using historical data on goods demand and the PSO-SVM method, it was possible to predict the future state of the goods market for 6 months. This method outperformed others in this regard, giving businesses a fresh idea for how to move forward with developing green innovation and a data economy based on IoT technology. In future research, improving the types of goods identification, enhancing the accuracy of identification, and taking into account more human and other external factors for prediction model building to improve its robustness will be the focus of work.

### Funding

This work is funded by the Plan Project of the Colleges' Science and Technology Innovation Team in Hubei (NO. T2021050) and the Science Research Project of Wuhan City College (NO. 2022CYYBKY05). The funders had no role in study design, data collection and analysis, decision to publish, or preparation of the manuscript.

### Grant Disclosures

The following grant information was disclosed by the authors:
Plan Project of the Colleges' Science and Technology Innovation Team in Hubei: T2021050.
Science Research Project of Wuhan City College: 2022CYYBKY05.

### Competing Interests

The authors declare that they have no competing interests.

### Author Contributions

- Ruilin Song conceived and designed the experiments, performed the experiments, analyzed the data, performed the computation work, authored or reviewed drafts of the article, and approved the final draft.
- Hui Hu conceived and designed the experiments, performed the experiments, analyzed the data, prepared figures and/or tables, authored or reviewed drafts of the article, and approved the final draft.

### Data Availability

The data and code are available in the Supplemental Files.

### Supplemental Information

Supplemental information for this article can be found online at http://dx.doi.org/10.7717/peerj-cs.1416#supplemental-information.

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
