# Peer review of "Impact of green technology innovation based on IoT and industrial supply chain on the promotion of enterprise digital economy"

_PeerJ Computer Science, doi:10.7717/peerj-cs.1416_

## Round 0.1 · original submission · Major Revisions

Dear authors

Your paper has be deeply examined by the experts in the relevant field, and myself. We feel that the paper has merits to consider but needs many changes which can improve the quality of your manuscript. Therefore, we invite you to incorporate these suggestions carefully and resubmit. Please also take this chance to further improve the language of your manuscript.
Thanks.

Reviewer 1 ·

Basic reporting

The development of a green economy has become the goal pursued by allcountries. As a trend of green innovation, digital economy has also become aresearch hotspot for scientists. In this paper, the problems of enterprisesupply chain management under the background of green innovation and digitaleconomy development are studied. Through the Internet of Things and artificialintelligence technology, the warehouse goods identification and demandprediction are completed. This paper is innovative and the writing logic isrigorous. My suggestion is to accept it after modification. The following arespecific suggestions for modificationl  The literature review should pay more attentionto the application of technical means.l  Lack of Yolov4 introduction and applicationscenario analysis;l  There is no necessary connection betweenSection 3.1 and Section 3.2, and the two parts seem to be separated;l  It is not necessary to introduce thecalculation methods of Precision, Recall and F1-score (formulas (6) ~ (8));l  If possible, it is better to present the dataset in a visual way;l  Avoid vague statements such as "Theprevious month's data of the company".l  There are also some problems in languageexpression in this paper, which need to be modified. l  Please check the Chinese characters in thereplacement formula and the redundant space characters in the references.

Experimental design

The development of a green economy has become the goal pursued by allcountries. As a trend of green innovation, digital economy has also become aresearch hotspot for scientists. In this paper, the problems of enterprisesupply chain management under the background of green innovation and digitaleconomy development are studied. Through the Internet of Things and artificialintelligence technology, the warehouse goods identification and demandprediction are completed. This paper is innovative and the writing logic isrigorous. My suggestion is to accept it after modification. The following arespecific suggestions for modificationl  The literature review should pay more attentionto the application of technical means.l  Lack of Yolov4 introduction and applicationscenario analysis;l  There is no necessary connection betweenSection 3.1 and Section 3.2, and the two parts seem to be separated;l  It is not necessary to introduce thecalculation methods of Precision, Recall and F1-score (formulas (6) ~ (8));l  If possible, it is better to present the dataset in a visual way;l  Avoid vague statements such as "Theprevious month's data of the company".l  There are also some problems in languageexpression in this paper, which need to be modified. l  Please check the Chinese characters in thereplacement formula and the redundant space characters in the references.

Validity of the findings

The development of a green economy has become the goal pursued by allcountries. As a trend of green innovation, digital economy has also become aresearch hotspot for scientists. In this paper, the problems of enterprisesupply chain management under the background of green innovation and digitaleconomy development are studied. Through the Internet of Things and artificialintelligence technology, the warehouse goods identification and demandprediction are completed. This paper is innovative and the writing logic isrigorous. My suggestion is to accept it after modification. The following arespecific suggestions for modificationl  The literature review should pay more attentionto the application of technical means.l  Lack of Yolov4 introduction and applicationscenario analysis;l  There is no necessary connection betweenSection 3.1 and Section 3.2, and the two parts seem to be separated;l  It is not necessary to introduce thecalculation methods of Precision, Recall and F1-score (formulas (6) ~ (8));l  If possible, it is better to present the dataset in a visual way;l  Avoid vague statements such as "Theprevious month's data of the company".l  There are also some problems in languageexpression in this paper, which need to be modified. l  Please check the Chinese characters in thereplacement formula and the redundant space characters in the references.

Additional comments

The development of a green economy has become the goal pursued by allcountries. As a trend of green innovation, digital economy has also become aresearch hotspot for scientists. In this paper, the problems of enterprisesupply chain management under the background of green innovation and digitaleconomy development are studied. Through the Internet of Things and artificialintelligence technology, the warehouse goods identification and demandprediction are completed. This paper is innovative and the writing logic isrigorous. My suggestion is to accept it after modification. The following arespecific suggestions for modificationl  The literature review should pay more attentionto the application of technical means.l  Lack of Yolov4 introduction and applicationscenario analysis;l  There is no necessary connection betweenSection 3.1 and Section 3.2, and the two parts seem to be separated;l  It is not necessary to introduce thecalculation methods of Precision, Recall and F1-score (formulas (6) ~ (8));l  If possible, it is better to present the dataset in a visual way;l  Avoid vague statements such as "Theprevious month's data of the company".l  There are also some problems in languageexpression in this paper, which need to be modified. l  Please check the Chinese characters in thereplacement formula and the redundant space characters in the references.

Reviewer 2 ·

Basic reporting

Green innovation and data economy complement each other, and technologies such as the Internet of Things and artificial intelligence, as intermediaries, need more application scenarios to unlock the potential of such technologies. Aiming at the inventory problem of enterprise supply chain, this paper presents a management framework which conforms to both green innovation concept and digital economy development, in order to develop new ideas for enterprises in the future.

After careful reading the authors need a revision of the paper. The presentation of the paper needs a professional improvement.

(1) Is there a correspondence between the identification of warehouse products by YOLOv4 and product requirements?
(2) Further consider modifying the title of the article. Elements such as "green technology innovation" and "Internet of Things" are not well reflected in the manuscript.
(3) Please note that acronyms of terms used just once in the abstract need not be included. Instead, the acronyms can be introduced in the main text, where they are repeatedly mentioned.
(4) The background analysis of the abstract is not enough. The contents of the main issues should be elaborated.
(5) The advantages of Internet of Things technology and its impact on supply chain development are not highlighted in the introduction; ,
(6) Add references to formulas (2) ~ (5);
(7) Please give more details about the theoretical foundations to support and further clarify these formulas.

Experimental design

(8) The description of the data set is inadequate. “The current application of IoT in warehouse management of supply chain is mainly video monitoring technology, so this paper selects the highest selling product in this warehouse in the last month for identification.”

Validity of the findings

(9) Why only the data from January to June are selected (Figure 6), which seems to be unable to accurately reflect the random seasonal variation trend of commodities.

Additional comments

(10) Revise the English thoroughly before submission.

Annotated reviews are not available for download in order to protect the identity of reviewers who chose to remain anonymous.

---

## Round 0.2 · accepted · Accept

Thanks for your fine contribution, and good luck for your future research.

Reviewer 1 ·

Basic reporting

This paper is well revised.

Experimental design

This paper is well revised.

Validity of the findings

This paper is well revised.

Additional comments

This paper is well revised.

Reviewer 2 ·

Basic reporting

The article can be accepted in its current version

Experimental design

The experimental design is sufficient

Validity of the findings

Validity of the findings is sufficient

Additional comments

The article can be accepted in its current version